# Vitamin E Blocks Connexin Hemichannels and Prevents Deleterious Effects of Glucocorticoid Treatment on Skeletal Muscles

**DOI:** 10.3390/ijms21114094

**Published:** 2020-06-08

**Authors:** Elisa Balboa, Fujiko Saavedra, Luis A. Cea, Valeria Ramírez, Rosalba Escamilla, Aníbal A. Vargas, Tomás Regueira, Juan C. Sáez

**Affiliations:** 1Departamento de Fisiología, Pontificia Universidad Católica de Chile, Santiago 8331150, Chile; fujiko.saavedra.l@gmail.com (F.S.); valeriapazramirez@gmail.com (V.R.); rescamhdz@yahoo.com (R.E.); anvargas@uc.cl (A.A.V.); 2Instituto de Ciencias Biomédicas, Facultad de Ciencias de la Salud, Universidad Autónoma de Chile, Santiago 8910060, Chile; luis.cea@uautonoma.cl; 3Instituto de Neurociencias, Centro Interdisciplinario de Neurociencias de Valparaíso, Universidad de Valparaíso, Valparaíso 2381850, Chile; 4Instituto de Ciencias de la Salud, Universidad de O’Higgins, Rancagua 2820000, Chile; 5Centro de Pacientes Críticos, Clínica las Condes, Santiago 7591047, Chile; tregueira@gmail.com

**Keywords:** connexons, mitochondrial dysfunction, oxidative stress, muscle atrophy, dexamethasone

## Abstract

Glucocorticoids are frequently used as anti-inflammatory and immunosuppressive agents. However, high doses and/or prolonged use induce undesired secondary effects such as muscular atrophy. Recently, de novo expression of connexin43 and connexin45 hemichannels (Cx43 HCs and Cx45 HCs, respectively) has been proposed to play a critical role in the mechanism underlying myofiber atrophy induced by dexamethasone (Dex: a synthetic glucocorticoid), but their involvement in specific muscle changes promoted by Dex remains poorly understood. Moreover, treatments that could prevent the undesired effects of glucocorticoids on skeletal muscles remain unknown. In the present work, a 7-day Dex treatment in adult mice was found to induce weight loss and skeletal muscle changes including expression of functional Cx43/Cx45 HCs, elevated atrogin immunoreactivity, atrophy, oxidative stress and mitochondrial dysfunction. All these undesired effects were absent in muscles of mice simultaneously treated with Dex and vitamin E (VitE). Moreover, VitE was found to rapidly inhibit the activity of Cx HCs in freshly isolated myofibers of Dex treated mice. Exposure to alkaline pH induced free radical generation only in HeLa cells expressing Cx43 or Cx45 where Ca^2+^ was present in the extracellular milieu, response that was prevented by VitE. Besides, VitE and two other anti-oxidant compounds, Tempol and Resveratrol, were found to inhibit Cx43 HCs in HeLa cells transfectants. Thus, we propose that in addition to their intrinsic anti-oxidant potency, some antioxidants could be used to reduce expression and/or opening of Cx HCs and consequently reduce the undesired effect of glucocorticoids on skeletal muscles.

## 1. Introduction

Glucocorticoids, such as dexamethasone (Dex), are widely used in clinic to treat diverse pathological conditions such as systemic sepsis, autoimmune, lymphoproliferative conditions and Duchenne muscular dystrophy. Despite their benefit as anti-inflammatory agents, it has been shown that glucocorticoids cause skeletal muscle atrophy [1,2] and activation of the inflammasome [3].

We have previously shown that Dex induces the rapid (<5 h) expression of connexin 43 and connexin 45 (Cx43 and Cx45, respectively) in mice skeletal muscles, leading to muscle atrophy [4]. Although the exact mechanism of glucocorticoids action on the expression of connexins (Cxs) in myofibers remains largely unknown, it is conceivable that they upregulate the protein and mRNA expression of Cx as demonstrated for Cx43 in activated stellate cells treated with Dex for four days [5]. Cxs are integral membrane proteins that form non selective channels in unopposed cell membrane, which are called hemichannels (HCs) or connexons. Currently, it is accepted that connexin hemichannels (Cx HCs) connect the intracellular and extracellular space, enabling the passage of ions as well as small nutrients (i.e., glucose), metabolites (i.e., reduced glutathione) and signaling molecules such as ATP, NAD^+^, cADPR, IP_3_, glutamate and prostaglandins E2 (reviewed in [6]).

During early and late stages of myogenesis, myoblasts express Cxs and form intercellular communicating junctions, called gap junctions that are essential for development. However, at the terminal stage of myogenesis the expression of Cxs is abrogated [7]. In fact, Cx proteins are no longer expressed a few days after birth when the skeletal muscles are innervated [6]. Thus, fully differentiated skeletal myofibers do not express Cxs. However, myofibers present Cxs re-expression during the regeneration process after damage or after denervation, as well as under inflammatory conditions [6,8]. Once Cxs are expressed, HCs can be inserted in the sarcolemma where upon opening can alter the electrochemical gradient across this membrane [9]. Consequently, the first changes so far detected correspond to a reduction in resting membrane potential and an increase of cytoplasmic Ca^2+^ signal [6]. In favor of these changes, the opening of Cx HCs can be enhanced by increase in cytoplasmic Ca^2+^ concentration and oxidative stress [10]. In addition, emerging evidences have led to propose a role for Cx HCs in the generation and maintenance of oxidative stress, revealing that cells expressing Cx43 had higher level of protein carbonyl modification [11]. Furthermore, open Cx HCs lead to loss of glutathione (GSH) [11], the major intracellular antioxidant agent. In support of the role of oxidative stress-induced opening of Cx HCs, dithiotreitol, a reducing agent of disulfide bounds, abrogates the increase in open probability of nitrosylated Cx HCs [12]. However, the effect of other potent antioxidant agents such as vitamin E (VitE) on Cx HCs remains unknown.

The involvement of Cx HCs has been demonstrated in a variety of pathological conditions associated with oxidative stress [10,13]. Interestingly, oxidative stress has been widely proposed as a mechanism that also triggers muscle atrophy [14] and the possible relationship between expression of Cx HCs and oxidative stress remains unknown.

Animal models of muscle atrophy commonly show increases in ROS in skeletal muscles, suggesting that ROS could play a role in the progression of muscle atrophy [15,16]. Accordingly, Dex induces ROS generation in muscle cells, which results in mitochondrial dysfunction [17]. Moreover, a 24-hours treatment of murine neural stem cells with Dex decreases the expression of ~72% of genes related to the respiratory chain, and alters the expression of ~43% of genes related to antioxidant activity. These evidences strongly suggest that Dex induces muscle atrophy by increasing the production of ROS. Studies have reported that oxidative stress can induce cell membrane depolarization [18] and the Cx HCs opening could participate in this phenomena [9]. The reduced electrochemical gradient across cell membrane might also affect negatively the normal function of metabolic pathways and organelles, including mitochondria. However, it is unclear which change occurs first. We speculated that oxidative stress and Cx HCs opening are key factors in inducing Dex-mediated muscle atrophy.

The aim of the present study was to investigate the effects of vitamin E (VitE), a potent antioxidant, on Dex-induced skeletal muscle atrophy in vivo. We found that Vit E blocks Cx HCs and prevents muscle atrophy, oxidative stress and mitochondrial dysfunction in myofibers of Dex-treated mice.

## 2. Results

### 2.1. Vitamin E Mitigates Dexamethasone-Induced Weight Loss and Muscle Atrophy in Mice

A growing body of evidence has shown that the prolonged use of Dex leads to muscle atrophy in mice [19,20]. There is also evidence that suggest a direct correlation between catabolic response and oxidative stress upon glucocorticoid treatment [21], which contributes to steroid induced-skeletal muscle atrophy [22]. Since VitE is generally considered a potent antioxidant, we analyzed its effects on Dex-induced skeletal muscle atrophy in mice.

In this model of Dex-induced muscle atrophy, VitE drastically reduced the damage induced by Dex. First, we analyzed body weight as indicative of muscle loss, which was evaluated during the full treatment period. Treatment with 10 mg/kg Dex induced a significant weight loss in mice compared to mice treated with saline, confirming that an animal model of Dex-induced muscle atrophy was successfully implemented (Figure 1A). Moreover, gastrocnemius muscles of Dex treated wild type mice were smaller than those of saline-treated mice (Appendix A). Notably, the VitE rich diet reduced the weight loss in mice co-treated with Dex compared to those treated only with Dex. Body weight started to decline on the second day of treatment in Dex-treated mice, and the decrease in body weight was approximately by 2 g compared to those mice fed with the control or VitE rich diet (Figure 1A).

To further investigate muscle loss, the cross sectional area (CSA) of myofibers from tibialis anterior (TA) muscles were measured. As expected, Dex caused a significant decrease (~20%) in the CSA of myofibers compared to myofibers of control mice. Moreover, the CSA of myofibers from muscles of mice co-treated with Dex and VitE were comparable to those of myofibers from control mice (3.1096 ± 146 vs 3.044 ± 159 µm^2^, *p* = 0.8199, respectively) (Figure 1B).

An immunohistochemistry analysis of atrophic muscles showed a greater number (~ 1.6 fold) of myofibers immune positive for atrogin-1, a muscle atrophy marker, in Dex-treated mice compared to control mice. In addition, atrogin-1 immunoreactivity was significantly less intense in myofibers from mice co-treated with Dex and VitE compared to mice treated only with Dex (Figure 1C).

### 2.2. Vitamin E Prevents the Dexamethasone-Induced Increase in Connexin Immunoreactivity of Skeletal Myofibers

Previously, we demonstrated that 5 h of Dex treatment induces de novo expression of Cx43 and Cx45 in skeletal muscles, and that the Dex treatment induces muscle atrophy [4]. Thus, we decided to analyze if Cx43/Cx45 proteins are detected in myofibers of mice treated with Dex for 7 days. We found that myofibers of TA muscles from Dex-treated mice presented high Cx43 and Cx45 immunoreactivity in their contour. Interestingly, such increase in Cx immunoreactivity was much lower in myofibers of mice co-treated with Dex and VitE, and absent in myofibers of mice treated with saline (Figure 2A,B).

As Dex induces the Cxs expression, we evaluated whether treatment with 10 mg/kg Dex induces muscular atrophy in Cx43/Cx45 expression-deficient (Cx43^fl/fl^Cx45^fl/fl^:Myo-Cre) mice. We found that although the mice do lose weight, they are resistant to developing muscular atrophy according to CSA, size of GS and atrogin-1 immunoreactivity (Appendix A).

### 2.3. Vitamin E-Rich Diet Prevents Oxidative Stress in Myofibers of Dexamethasone-Treated Mice, and Connexin Expression is Necessary for Dexamethasone to Induce Oxidative Stress

To analyze if Dex induces oxidative stress in this animal model, and if VitE decreases oxidative stress, we measured ROS production in TA muscles using the dihydroethidium (DHE) probe, which reacts with O_2_^−.^ to form ethidium that can be visualized in a fluorescent field [23]. A significant increase (~2.5 fold) in DHE fluorescence intensity was observed in myofibers of TA muscles of Dex-treated mice compared to myofibers from control mice. The fluorescence intensity was significantly lower in myofibers from mice co-treated with Dex and VitE (Figure 3A).

We previously demonstrated that Dex induces the expression of Cxs in myofibers of skeletal muscles within 5 h [4]. Now, we hypothesized that the appearance of Cxs in the sarcolema of myofibers occurs prior to the onset of oxidative stress. To test this hypothesis, we measured oxidative stress in TA muscles from Cx43/Cx45 expression-deficient (Cx43^fl/fl^Cx45^fl/fl^:Myo-Cre) skeletal myofibers mice treated with Dex. We found that these myofibers presented similar DHE fluorescence to that of myofibers from control mice (mean 0.160 ± 0.01 vs. 0.180 ± 0.01, respectively) (Figure 3B), suggesting that Cx expression precedes and is needed for the induction of oxidative stress by Dex.

### 2.4. Mitochondrial Dysfunction Induced by Dexamethasone Requires the Expression of Connexins and is Prevented by Vitamin E

Disturbed redox signaling due to greater production of ROS is an important regulator of cell signaling pathways that control both proteolysis and protein synthesis in skeletal muscle [24,25,26], and oxidative stress is a known contributing factor in mitochondrial dysfunction. Given our previous results, we decided to evaluate the effects of Dex and VitE on mitochondrial function. To assess mitochondrial function, we evaluated mitochondrial oxygen consumption rates (OCRs), mitochondrial membrane potentials (MMPs) and mitochondrial superoxide production (mSO).

OCRs in skeletal myofibers of treated mice were evaluated in freshly isolated and permeabilized myofibers. Respiration was stimulated by the addition of substrates: glutamate and malate (Glu/Mal, state 2, complex I), saturating ADP (ADP, state 3, complex I), succinate (Succ, state 3 complexes I + II), or suppressed by the addition of the inhibitors rotenone (Rot, state 3 complex II) and antimycin A (AntA, background). A clear decrease in OCR was recorded in myofibers from mice treated with Dex. However, this was not seen in myofibers from mice co-treated with Dex and VitE that behaved similar to control myofibers (Figure 4A). In addition, we evaluated mitochondrial membrane potential (MMP) with Mitotracker Red (MtRed), which is a red-fluorescent dye that stains mitochondria in live cells, and its accumulation depends on membrane potential. MMP in muscles of Dex-treated mice was lower than in myofibers from control mice. Again, MMP of myofibers from mice co-treated with Dex and VitE was similar to that of myofibers of control mice (Figure 4B).

mSO generation was evaluated with Mitosox, which is a fluorogenic dye widely used to detect mitochondrial ROS, especially O_2_^-.^ in live cells, and whose oxidation produces red fluorescence. This probe revealed that fluorescence intensity was stronger in myofibers of Dex-treated mice compared to myofibers from control mice, and myofibers from mice co-treated with Dex and VitE (Figure 4C), suggesting that the production of ROS in the mitochondria of myofibers from Dex-treated mice and VitE prevents it.

As we previously saw, Cx expression is necessary for the induction of ROS in muscle from mice treated with Dex. We also wanted to evaluate mitochondrial function in Cx43/Cx45 expression-deficient mice to explore whether Cxs contribute to the progression of mitochondrial dysfunction. We found no differences in OCR between Cx43/Cx45 expression-deficient myofibers from mice treated with Dex or saline (Figure 4D). We also did not find significant differences in MMP (Figure 4E) and mSO (Figure 4F) between Cx43/Cx45 expression-deficient myofibers from mice treated with saline solution and myofibers from mice treated with Dex (Figure 4E,F). The differences in OCR between the wild type mice (Figure 4A) and Cx43^fl/fl^Cx45^fl/fl^:Myo-Cre mice (Figure 4D) could be explained because the last strain is on a genetic background different from that wild type mice, but what we want to highlight is that the Cx43^fl/fl^Cx45^fl/ fl^:Myo-Cre mice do not present mitochondrial dysfunction associated with Dex treatment. These results suggest that Cx expression is also necessary for the development of mitochondrial dysfunction induced by Dex.

### 2.5. Vitamin E Blocks the connexin hemichannel activity

In order to evaluate the functional state of Cx HCs, we performed dye uptake measurements in freshly isolated myofibers from TA muscles from saline- and Dex-treated mice. Etd^+^ uptake, mainly reflecting Cx HC activity was significantly higher in myofibers from Dex-treated mice (Figure 5A) compared to myofibers from control mice. Whereas myofibers from mice treated in conjunction with Dex and VitE exhibited lower dye uptake as compared to that of myofibers from Dex-treated mice (Figure 5A). Etd^+^ uptake of myofibers was partially blocked by La^3+^, a Cx HC and P2X receptor inhibitor [27], which suggest that in addition to Cx channels there could be other non-selective channels involved in the increased dye uptake observed.

To test if VitE could directly inhibit Cx HCs, we measured Etd^+^ uptake in HeLa-Parental cells, which do not express Cx HCs, as well as in HeLa-Cx43EGP and HeLa-Cx45 cells, which express Cx43EGP and Cx45 HCs, respectively. Etd^+^ uptake was measured for 5 min in cells exposed to a saline solution with physiologic extracellular divalent cation concentrations (basal in Figure 5B). In order to increase the open probability of Cx HCs, cells were bathed in Ca^2+^/Mg^2+^ free solution as reported previously [28,29]. A rapid increase in Etd^+^ uptake was recorded in HeLa-Cx43EGP and -Cx45 cells, but not in HeLa-Parental cells (Ca^2+^/Mg^2+^-free in Figure 5B). Lastly, 300 μM VitE added to the bath solution reduced Etd^+^ uptake in HeLa-Cx43EGP and -Cx45 cells induced by free Ca^2+^/Mg^2+^ extracellular saline to values similar to those recorded in the presence of physiologic concentrations of extracellular divalent cations (VitE 300 μM in Figure 5B). Then, Etd^+^ uptake was drastically inhibited by La^3+^, a widely used Cx HC blocker (Figure 5B). Taken together, these results suggest that VitE inhibits the activity of Cx43 and Cx45 HCs in myofibers and in HeLa cells.

### 2.6. Opening of Connexin Hemichannels Promotes Generation of Reactive Oxygen Substances in HeLa Cells

In order to identify the mechanism by which oxidative damage is associated with the opening of Cx HCs, we tested whether opening of Cx HCs could underlie the observed increase in ROS generation. To evaluate the amount of ROS in cells we used the DHE probe, which enters to the nuclei when oxidized and fluoresce red. The cells were exposed to a Krebs solution of physiological pH (control), a divalent cation-free solution (Ca^2+^/Mg^2+^-free), an alkaline solution (pH 8.4) or an alkaline solution plus VitE. Divalent cation-free as well as alkaline solution are known to increase the open probability of hemichannels formed by these Cxs [30,31]. Out of these conditions, HeLa cells expressing Cx43 and Cx45 exposed to alkaline solution were positive for DHE staining in the nuclei (Figure 6), indicating an increase in ROS generation. This response was absent in cells that did not express Cxs (PAR), and in cells exposed to divalent cation-free solution that does not have Ca^2+^ (Figure 6), but favors opening of Cx HCs [9,30]. Treatment with VitE prevented the appearance of ROS in HeLa Cx43 and Cx45 cells exposed to alkaline solution (Figure 6). These results suggest that open Cx HCs favor the generation of ROS, possibly triggered by the Ca^2+^ influx via these membrane channels.

### 2.7. The Antioxidants Tempol and Resveratrol but Not N-acetylcysteine Block Connexin Hemichannel Activity

Previously, it has been shown that some antioxidant molecules prevent muscle atrophy induced by conditions that induce the expression of Cx HS in skeletal myofibers such as denervation and myopathy of mdx mouse, a model of Duchenne muscular dystrophy [32]. However, the cause–effect of ROS generation and Cx HC activity in skeletal muscles of animal model treated with glucocorticoids has not been established. Therefore, we decided to study the possible effect of three antioxidants on Cx43 HC activity by measuring the Etd^+^ uptake of HeLa-Cx43 cells exposed to a divalent cation-free solution to increase the open probability of Cx HCs. The increase in Etd^+^ uptake response (~3.5 fold) was completely blocked by carbenoxolone (Cbx, 200 μM), which is a well-known Cx HC blocker (Figure 7)**,** indicating that Etd^+^ uptake occurred through open HCs. Interestingly, the antioxidants Tempol (500 μM) and Resveratrol (200 μM) also drastically reduced the Etd^+^ uptake rate (~3.6 and ~3.3 fold, respectively). However, high NAC concentration (5 mM) produced only a small reduction in Etd^+^ uptake rate (Figure 7). These results suggest the participation of Cx HCs in the induction of ROS and that some antioxidants can exert their antioxidant action by reducing the activity of Cx HCs, and therefore the entry of Ca^2+^ into the cell.

## 3. Discussion

In the present work, we showed that VitE protects against the deleterious effect caused by Dex treatment on skeletal muscles by blocking Cx HCs. Our results indicate that inhibition of Cx HCs by VitE is associated with the prevention of muscle mass loss, mitochondrial dysfunction and oxidative damage in myofibers from Dex-treated mice (Figure 8). These findings suggest that the pharmacological inhibition of Cx HCs may be a powerful intervention to prevent muscle atrophy induced by long term treatment with glucocorticoids.

Progressive weight loss was found in mice treated with Dex, which could be due to several factors including dehydration and skeletal muscle atrophy. Since skeletal muscles correspond to about 50% of the body weight of a mouse [33], it is likely that Dex-induced muscle atrophy played a relevant role in the total weight loss. In agreement with the latter interpretation, a significant reduction in CSA of myofibers from TA muscles (~20%) was observed in Dex-treated mice. Moreover, myofibers of Dex treated mice showed a significant increase in immunoreactivity of atrogin, an important regulator of ubiquitin-mediated protein degradation in skeletal muscle [34]. Besides body weight and CSA of myofibers from TA muscles remained normal in mice treated with Dex plus VitE, which suggest muscle atrophy prevention by vitamin E. Consistent with the latter, we found that VitE drastically prevented Dex-induced activation of Atrogin-1.

In agreement with the hypothesis that ROS-related signaling pathways are involved in skeletal muscle atrophy [26], we found a significantly high DHE signal in myofibers of Dex-treated mice. Along the same line of analysis, an antioxidant therapy should be effective in reducing Dex-induced skeletal muscle atrophy. Accordingly, we found that VitE, the most lipid-soluble component in the cell antioxidant defense system [35], drastically reduced the DHE signal generation of ROS in skeletal myofibers of Dex-treated mice. This decrease in ROS can be attributed mainly to the following two mechanisms exerted by VitE: 1) preventing an upstream step of ROS production; and 2) scavenging ROS as antioxidant defense. In support of the first possibility, it is likely that VitE prevented the generation of ROS by inhibiting Cx expression and Cx HC activity. In accordance, Dex treatment did not induce ROS generation in myofibers deficient in Cx43 and Cx45 expression. The above interpretation is also in agreement with our findings since myofibers of mice simultaneous treated with VitE and Dex did not present significant Cx43 and Cx45 immunoreactivity and the acute application of VitE blocked Cx HCs in freshly isolated myofibers of Dex treated mice or HeLa cells transfected with Cxs and exposed to divalent cation solution. Similar results have been previously obtained in Cx43 and Cx45 expression during endotoxemia and treatment with boldine that rapidly blocked Cxs and completely abrogated Cx45 and Cx43 immunoreactivity [36].

Since skeletal myofibers express the mRNAs of Cxs 43 and 45 [37], the preventive effect of VitE on Dex-induced Cx expression might be explained by direct inhibition of mRNA translation. Moreover, open Cx HCs might allow the intracellular activation of a signal that promotes Cx translation. In support of this interpretation it has been previously reported that LLC-PK1 cells transfected with Cx43 and treated with Cd^2+^, which induces opening of Cx43 HCs, leads to activation of Cx43 expression via a JNK-dependent pathway [38]. Future studies are required to demonstrate whether VitE affects this cell signaling pathway mediated by Cx HCs. It has been shown that oxidant agents induce opening of Cx HCs [12,39]. H₂O₂ increases hemichannel function and Cx43 expression on the cell surface [40]. It is possible that VitE may also inhibit the expression of Cxs by decreasing ROS.

Previous studies have shown that ROS increase intracellular Ca^2+^ and elevate cytoplasmic Ca^2+^ concentration, as well as reduce redox potential and enhance the activity and amount of Cx HCs in the cell membrane [12,28]. At least for Cx43 HCs, the mechanism involves nitrosylation of Cx43 and the effect is reversed by a sulfhydryl reducing agent such as the DTT [12]. However, we found that concentration of Tempol, Resveratrol and VitE, frequently used as antioxidants in biological systems [41,42], rapidly and drastically reduced the Cx HC permeability to Etd^+^ in the absence of a redox stress, since cells were exposed to divalent cation-free solution to increase the open probability of HCs, suggesting that their inhibitory effect on Etd^+^ uptake is unrelated to redox changes and is more likely due to direct blockade of the HCs. Interestingly, the very high concentration (5 mM) of the NAC used in this study caused only a tendency to reduce the Etd^+^ uptake (not statistically significant). Since NAC compared to Tempol, Resveratrol and VitE, is more flexible and with smaller molecular diameter, enabling it to permeate the HCs and compete with Etd^+^ in the dye uptake assay, as demonstrated for other HC permeant molecules, and reduces the Etd^+^ uptake rate [29] but does not necessary act as a Cx HC blocker. Clearly, more studies will be needed to clarify these issues. The unexpected effect of Tempol, Resveratrol and VitE as inhibitors of Cx43 HCs leads to propose that in addition to their direct and potent antioxidant property they could drastically reduce the generation of ROS by hindering the Ca^2+^ influx via Cx43 HCs. In agreement with this idea it has been shown that Cx43 HCs are permeable to Ca^2+^ [31] and this divalent cation is required for the activation of several intracellular metabolic pathways that generate ROS [43].

We have previously shown that Cx HCs expressed by myofibers exposed to Dex enable the increase in basal intracellular Ca^2+^ signal [4]. This persistent increase in cytoplasmic Ca^2+^ is a versatile and ubiquitous cellular signal that activates catabolic metabolism [44,45,46]. Given that Cx43 and Cx45 HCs are permeable to Ca^2+^ [30,47], their presence in the sarcolemma of myofibers from Dex treated mice should favor the influx of Ca^2+^ [47]. Thus, we speculate that the Ca^2+^ signal due to Ca^2+^ influx through Cx HCs would be a critical cause of the Dex-induced myofibers damage. In support of this possibility, it has been shown that an increase in intracellular free Ca^2+^ concentration could lead to ROS generation [48]. In line with this interpretation, our in vitro results in HeLa cells that express Cx43 or Cx45 showed that under conditions in which open Cx HCs (but in the absence of extracellular Ca^2+^) there is no increase in ROS as quantified by DHE. However, Ca^2+^ entered the cells and increased ROS generation when cells were exposed to an alkaline environment containing Ca^2+^, which induces Cx HC opening as previously demonstrated [30] and a greater Etd^+^ uptake. In addition to generating oxidative stress, Ca^2+^ can activate other pathways that lead to muscle atrophy such as Ca^2+^-dependent and lysosomal protein breakdown [49]. Moreover, elevated cytoplasmic Ca^2+^ could affect the mitochondrial function. It is currently accepted that mitochondrial dysfunction plays a critical role in the development of muscle atrophy [50]. Alterations in mitochondrial morphology and function are frequently associated with skeletal muscle atrophy in many conditions such as different neuromuscular disorders [51]. Other muscle atrophy models indicate that mitochondria appear to be the dominant site for ROS production [52,53]. After Dex treatment, it has been observed that mitochondrial respiration is compromised before MuRF1 and Atrogin-1 muscle atrophy markers begin to increase [54]. In addition, muscle atrophy is prevented in animal models exposed to Dex only by improving mitochondrial function [55]. Based on such evidence and our results in Cx43/Cx45 expression-deficient myofibers, in which the increase in ROS production and mitochondrial dysfunction induced by Dex is absent, we propose that Cx HC activation due to Dex as well as the consequent disturbance of the electrochemical gradient across the sarcolemma (including the entry of Ca^2+^ into the cell) affect mitochondrial function increasing intracellular ROS levels. In support of this possibility, it has been demonstrated that elevated intracellular free Ca^2+^ concentrations reduce MMP, and could lead to greater generations of ROS as well as a stronger activation of the permeability transition pore, which contributes to reducing MMP [48].

Previously, it has been shown that Dex treatment causes intracellular ATP depletion, and Resveratrol efficiently reverses Dex-induced mitochondrial dysfunction as well as muscle atrophy [54]. These findings support our hypothesis that Cx HCs contribute to the development of Dex-induced atrophy, since greater intracellular free Ca^2+^ concentration activates both Cx HCs and Panx1 channels. Cx HCs and Panx1 channels enable the release of ATP to the extracellular milieu [47]. Interestingly, we found that Resveratrol and Tempol, two antioxidant agents frequently used for scavenging free radicals also inhibit Cx43 HCs in vitro.

In conclusion, our study showed that Dex induces the expression of functional Cx HCs in the sarcolemma of muscle cells. The early expression of Cx HCs and the subsequent entrance of Ca^2+^ would lead to mitochondrial dysfunction, which may mediate an increase in ROS, oxidative damage and muscle atrophy. Therefore, the use of VitE and possibly Resveratrol and Tempol might reduce muscle atrophy caused by long term treatment with glucocorticoids.

## 4. Materials and Methods

### 4.1. Reagents

Injectable dexamethasone (Dex: 4 mg/mL) was purchased from Laboratorio Sanderson S.A (Santiago, Chile). HEPES, N-benzyl-ptoluene (BTS), sulfonamide, collagenase type I, α-tocopherol acetate (Vitamin E: VitE), suramin, ethidium bromide (Etd^+^), n-benzyl p-toluenesulfonamide (BTS), Tempol, Resveratrol and N-acetylcysteine (NAC) were obtained from Sigma-Aldrich (St. Louis, MO, USA). DMEM/F12 culture medium, fetal bovine serum albumin (FBS), DHE, Mitosox, MitoTracker red and MitoTracker green were purchased from Thermo Fisher Scientific ( Waltham, MA, USA). The anti-atrogin-1 antibody was purchased from Abcam (ab 168372). Previously characterized anti-Cx43 and anti-Cx45 antibodies were used [56]. Cy2- or Cy3-conjugated goat anti-rabbit IgGs were purchased from Jackson Immuno Research (Indianapolis, IN, USA). DL-alpha tocopherol acetate (VitE) supplemented diet (Prolab 5P00 w/2000 IU/kg vitamin E (5AU8)) was obtained from Prolab Labdiet, St. Louis, MO, USA.

### 4.2. Animals

The protocols used in the present study were approved (August 2015) by the bioethics committee of Pontificia Universidad Católica de Chile (protocol N° 150515010). Eight weeks old C57BL6 wild-type (WT) male mice and as previously described [8] male mice that had skeletal myofibers-deficient for Cx43 and Cx45 expression (Cx43^fl/fl^Cx45^fl/fl^:Myo-Cre) were kept in standard housing conditions with a 12-hour:12-hour light-dark cycle with food and water ad libitum.

Mice from one group were injected (i.p.) daily for seven days with Dex (10 mg/kg). Another group of animals received Dex and VitE supplemented diet (2000 IU/kg DL-alpha tocopherol acetate) for seven days. Mice from control group received saline injections and control diet. Their body weights were evaluated daily during the full treatment period.

### 4.3. Cell Cultures

HeLa-Parental cells were obtained from ATCC (CCL-2; ATCC, Rockville, MD, USA). Previously described [57] HeLa cells stably transfected with mouse Cx43 (HeLa-Cx43), or Cx45 (HeLa-Cx45) cDNA were kindly provided by Dr. Klaus Willecke (Bonn University, Germany). All cell lines were grown at 37 °C and 5% CO_2_ in DMEM, supplemented with 10% FBS (GIBCO, Invitrogen), 100 U/mL penicillin, 100 μg/mL streptomycin sulfate, and 0.5 μg/mL puromycin to select transfected cells. Untransfected HeLa-Parental cells were used as controls.

### 4.4. Histological Analysis

For immunofluorescence or cross-sectional area (CSA) analysis of skeletal myofibers, tibialis anterior (TA) muscles were dissected, embedded in OCT (Andes import, Santiago, Chile) and fast frozen in liquid-nitrogen-cooled isopentane (Merck, Darmstadt, Germany). Cross sections of 16 µm thickness were obtained using a cryostat (Cryostat Leica, CM 100-1), placed on glass slides (B&C, Germany) and fixed (10 min) with 4% paraformaldehyde (Electron Microscopy Sciences, PA, USA). The CSA of skeletal muscle fibers was measured as previously described [8] and the CSA was evaluated by using offline analyses with ImageJ software (National Institutes of Health., Bethesda, MD, USA) [58].

### 4.5. Immunofluorescence

Cross sections of TA muscles obtained as described above were blocked (1 h) in blocking solution and processed as described previously [58] for detection of atrogin, Cx43 or Cx45. Antigen sites recognized by each antibody were detected with appropriate dilutions of Cy2- or Cy3-conjugated goat anti-rabbit IgGs (1:1000). Then, samples were washed 4 times with PBS 1X and once with distilled water, after which they were mounted with Fluoromount-G™ and nuclei were stained with DAPI (Electron Microscopy Science, Hatfield, PA, USA) and mounted on glass slides. The images were acquired in a Nikon Eclipse Ti microscope, and fluorescence intensity quantification was carried out using ImageJ. 10 images for each sample of at least 3 animals were analyzed per condition.

### 4.6. Skeletal Myofiber Isolation

Myofibers of mouse flexor digitorum brevis (FDB) muscles were isolated from anesthetized and sacrificed mice as previously described [56]. In brief, muscles were incubated for 2.5 h at 37 °C in dissociation medium (DMEM/F12, supplemented with 10% fetal bovine serum, and 0.2% collagenase type I). Then, muscles were transferred to a tube containing Krebs buffer plus 10 µM n-benzyl p-toluenesulfonamide, BTS (a contraction inhibitor, to reduce muscle damage). Then, muscle tissue was incubated for 2 min with 200 µM suramin to prevent the effects of ATP released damaged myofibers during the procedure [56] followed by gentle trituration through a wide-tip Pasteur pipette. Myofibers were pelleted by centrifugation at 1,000 rpm for 10 s (Kubota 8700 centrifuge, Tokyo, Japan) and washed with HEPES buffered Krebs saline solution containing 10 µM BTS. Then, partial dissociated myofibers were gently triturated by passing them through a narrow-tip Pasteur pipette in order to obtain completely dissociated myofibers. Finally, myofibers were centrifuged and resuspended in HEPES-buffered Krebs saline solution containing 10 µM BTS, and then placed on ice until further analyses.

### 4.7. Dye Uptake Assay

Cellular uptake of ethidium (Etd^+^) was evaluated in freshly isolated myofibers and HeLa cells by time-lapse measurements as described previously [39,56]. In brief, myofibers plated onto glass coverslips were washed twice with recording solution and incubated in 5 μM Etd^+^. The basal fluorescence intensity value was recorded for 5 min in regions of interest (nuclei) in different cells followed by 5 min of recording after the application of an HC blocker. Fluorescence intensity recording were performed using a Nikon Eclipse Ti inverted microscope, and NIS-Elements software (Nikon, Tokio, Japan).

In HeLa cells transfected with Cx43 the Etd^+^ fluorescence intensities of the nuclei were first recorded under basal conditions, followed by 3 washed with (Ca^2+^/Mg^2+^-free) Krebs and recorded for 5 min. Then, cells were treated with different antioxidants [Tempol (500 µM), Resveratrol (200 µM), NAC (5 mM) or Vit E (300 µM)] and recorded for additional 5 min to test their effect. At the end of each experiment, the Cx HC blocker La^3+^ was added to confirm the involvement of HC-mediated Etd^+^ uptake.

### 4.8. Laser Confocal Imaging

The mitochondrial function was evaluated in FDB myofibers loaded with MitoTracker Red CMXRos (25 nM) to determine mitochondrial membrane potential (MMP) or Mitosox (125 nM) a superoxide indicator or in Krebs buffer at 37 °C for 30 min. Afterward, cells were washed with Krebs for live imaging. Under both conditions cells were counterstained with MitoTracker Green (250 nM). Images were obtained using a NIKON laser scanning confocal microscope and fluorescence intensities were evaluated with ImageJ software (NIH, Bethesda, MD, USA). We used various concentrations of MitoSOX and we choose 125 nM because higher concentrations (e.g., 5 μM) of this probe may adversely affect the mitochondrial electron transport chain and lead to diffusion of MitoSOX from the mitochondrial compartment into the cytosol [59]. While the optimal concentration of MitoSOX for detecting mitochondrial ROS remains to be further determined, concentrations bellow 1 μM of MitoSOX appears more reliably to detect relative differences in mitochondrial ROS formation between control and mitochondrial DNA-deficient cells [60].

### 4.9. Oxygen Consumption

The oxygen consumption of muscle fibers was measured in fiber bundles from soleus muscle (0.2–0.8 mg dry weight), which were separated as described previously [58]. Then, bundles were permeabilized on ice with 50 µg/mL saponin for 30 min, and washed in MIR05 (in mM: 0.5 EGTA, 3 MgCl_2_-6H_2_O, 60 K-lactobionate, 20 taurine, 10 KH2PO_4_, 20 HEPES, 110 sucrose, 1 g/L BSA, pH 7.1). High-resolution O_2_ consumption measurements were performed at 30 °C with the OROBOROS Oxygraph-2K (OROBOROS Instruments, Innsbruck, Austria). The media and chemicals were prepared as described [61]. The oxygen flow in state II attributable to complex I was measured after the addition of glutamate (final concentration 10 mM) and malate (2 mM), followed by measure of oxygen flow in state III. Then, saturated ADP (final concentration: 5–10 mM) was added and then oxidation of complex I succinate (10 mM) was evaluated. Finally, electron transport through complex I and III was inhibited by the sequential addition of rotenone (0.1 μM) and antimycin A (10 μM), respectively, to obtain the oxygen flow independent of the electron transfer system.

### 4.10. Detection of ROS with DHE

To detect ROS, 10-μm thick fresh muscle sections were obtained using a cryostat microtome (Leica CM1100) and mounted onto glass slides. Slides were incubated (30 min at 37 °C) with 10 μM dihydroethidium (DHE) in PBS. DHE produces red fluorescence when oxidized to ethidium bromide by ROS, including superoxide anion [23]. After staining, sections were rinsed, air dried, mounted in Fluoromount-G™ (Electron Microscopy Science, Hatfield, PA, USA) and coverslipped.

HeLa cells were incubated for 10 min at 37 °C with 5 μM DHE in phosphate-buffered saline, then washed 3 times with PBS, fixed with 4% (wt/vol) paraformaldehyde for 5 min and mounted in Fluoromount-G™. All samples were examined under an epifluorescence microscope Nikon Eclipse Ti with a 20X epifluorescence objective.

### 4.11. Statistics

Results for each condition are presented in column bar graphs and each value corresponds to the means ± standard error of the mean (SEM). The number of mice per condition is indicated in the corresponding figure legend. The results were subjected to Student´s t test (two group comparison) and one/two-way ANOVA test (multiple comparisons) followed by Bonferroni as post-hoc tests. All analyses were performed using GRAPHPAD software. Significantly differences between groups were considered when *p* < 0.001; *p* < 0.01 or *p* < 0.05 as indicated in figure legends.

## Figures and Tables

**Figure 1 ijms-21-04094-f001:**
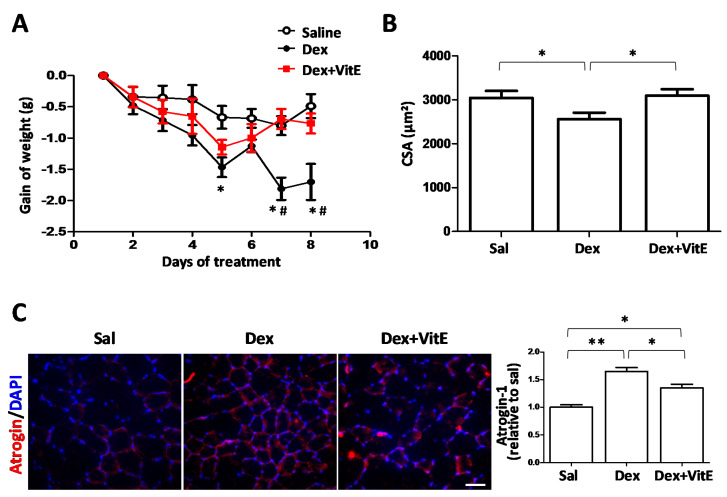
Effect of long term treatment with dexamethasone and vitamin E on weight loss and muscle atrophy in mice. Mice were daily treated for 7 days with saline (Sal) or dexamethasone (Dex; 10 mg/kg/day) and fed with vitamin E (VitE) supplemented diet (Dex + VitE). (**A**) The mice weight was recorded daily. The graph shows the loss of weight relative to the start of treatment. (Sal; n = 10; n Dex = 10; n Dex + VitE = 10). Significance was assessed by two-way ANOVA followed by Bonferroni post hoc test. * *p* < 0.05 Saline vs Dex; ^#^
*p* < 0.05 Dex vs Dex+VitE. (**B**) The cross-sectional area (CSA) of myofibers of tibialis anterior (TA) muscles was measured by off-line analysis of hematoxylin-eosin images. Five images of each muscle section were evaluated (Sal; n = 3; Dex; n = 3; Dex+ VitE; n = 3). The results are expressed as mean ± SEM. (**C**) The presence and cellular distribution of the protein-degradation marker Atrogin-1 was evaluated by immunofluorescence assay in cross sections of TA muscles. The graph shows the red fluorescence intensity measured using Image J software. The results are expressed as mean ± SEM.* *p* < 0.05; ** *p* < 0.01; comparing as indicated with brackets (n Saline: 5; n Dex: 5; n Dex+VitE: 3). Scale bar: 50 μm.

**Figure 2 ijms-21-04094-f002:**
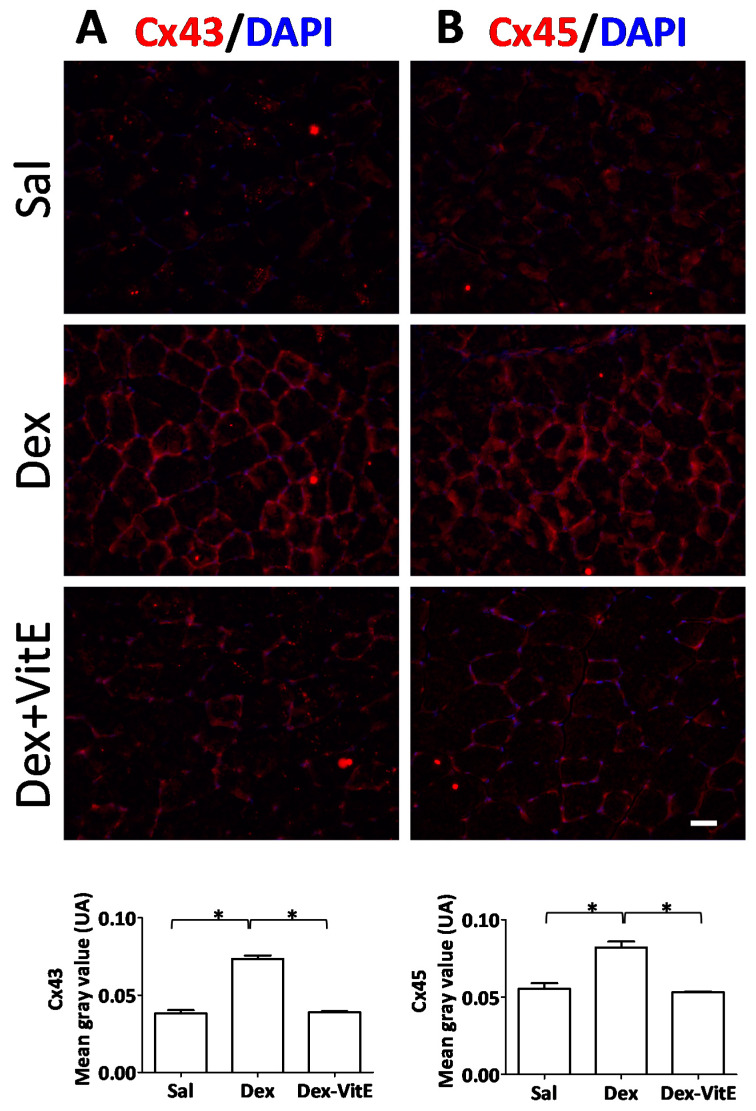
Vitamin E prevents the increase in connexin43 and Cx45 immunoreactivity induced by dexamethasone. The presence and cellular distribution of conneixn43 (Cx43, **A**) and conneixn45 (Cx45, **B**) was evaluated by immunofluorescence assays in slices of tibialis anterior muscle from mice treated with dexamethasone (Dex) or with saline solution or with Dex plus VitE rich diet (Dex + VitE). The graphs show the red fluorescence intensity for each connexin measured using Image J software. The results are expressed as mean ± SEM.* *p* < 0.05; comparing as indicated with brackets. Scale bar: 50 μm. (*n* = 3 individual animals per group).

**Figure 3 ijms-21-04094-f003:**
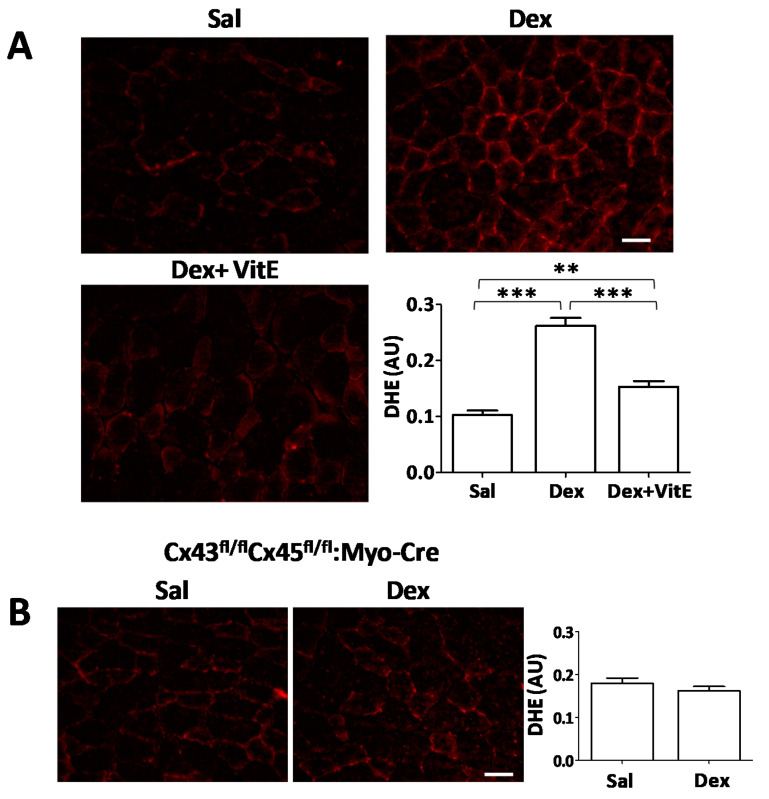
Vitamin E reduces the generation of reactive oxygen species induced by long term treatment with Dexamethasone. (**A**) Mice were treated daily during 7 days with saline (Sal) or dexamethasone (Dex; 10 mg/kg/day) or Dex and fed with Vitamin E (VitE) supplemented diet (Dex + VitE). Then, the oxidation of dihydroxy ethidium (DHE) was visualized in cross sections of TA muscles (red). The graph shows the red fluorescence intensity measured using Image J software. The results are expressed as mean ± SEM. (Saline; n = 5; Dex; n = 5; Dex+VitE; n = 3). (**B**) DHE staining in TA muscles from Cx43/Cx45 expression-deficient (Cx43^fl/fl^Cx45^fl/fl^:Myo-Cre) skeletal myofibers mice treated with Dex. The graph shows the red fluorescence intensity measured using Image J software. The results are expressed as mean ± SEM. (Sal; n= 5; Dex; n = 5). ** *p* < 0.01; *** *p* < 0.001; as indicated with brackets. Scale bar: 50 μm.

**Figure 4 ijms-21-04094-f004:**
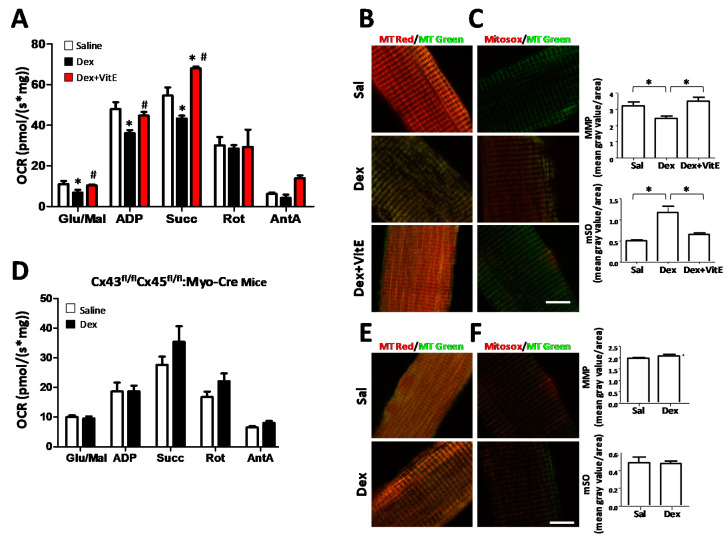
Vitamin E prevents the mitochondrial dysfunction generated by long term treatment with dexamethasone. Mice were daily treated for 7 days with saline (Sal) or dexametahasone (Dex; 10 mg/kg/day) or Dex and fed with Vitamin E (VitE) supplemented diet (Dex + VitE). (**A**) The myofibers were isolated from soleus muscles. The oxygen consumption rate (OCR) measurements were carried out on permeabilized muscle fibers by stimulating mitochondrial enzyme complexes in sequence. Respiration was stimulated by the addition of the following substrates: glutamate and malate (Glu/Mal, state 2, complex I), ADP (state 3, complex I), succinate (Succ, state 3 complex I + II), or in the presence of the inhibitors rotenone (Rot, state 3 complex II), antimycin A (AntA, background). The graph shows the average rate of oxygen consumption rate (OCR) for each condition. OCR was measured using the O2k system (OROBOROS Instruments). Values are means ± SEM (for each group *n* =5). * means significantly different from Saline *p* < 0.05; # means significantly different from Dex *p* < 0.05 (**B**) To measure the mitochondrial membrane potential (MMP), cells were loaded with MitoTracker Red CMXRos (red). Mitotracker green was used to counterstain mitochondria and are shown in green. The graph shows the red fluorescence intensity values using Image J software. a means significantly different from Saline *p* < 0.05; b means significantly different from Dex *p* < 0.05. (**C**) To measure mSO production in the mitochondrial matrix and stain mitochondria, the myofibers were stained with MitoSOX Red and Mitotracker Green, respectively. The graph shows the red fluorescence intensity measured using Image J software. a means significantly different from saline *p* < 0.05; b means significantly different from Dex *p* < 0.05. (**D**) OCR from Cx43/Cx45 expression-deficient (Cx43^fl/fl^Cx45^fl/fl^:Myo-Cre) skeletal myofibers mice treated with Dex (for each group *n* = 4). (**E**) MMP from Cx43/Cx45 expression-deficient (Cx43fl/flCx45fl/fl:Myo-Cre) skeletal myofibers mice treated with Dex (for each group *n* = 3). (**F**) mSO from Cx43/Cx45 expression-deficient (Cx43^fl/fl^Cx45^fl/fl^:Myo-Cre) skeletal myofibers mice treated with Dex (for each group *n* = 3). * *p* < 0.05; as indicated with brackets. Scale bar: 10 μm.

**Figure 5 ijms-21-04094-f005:**
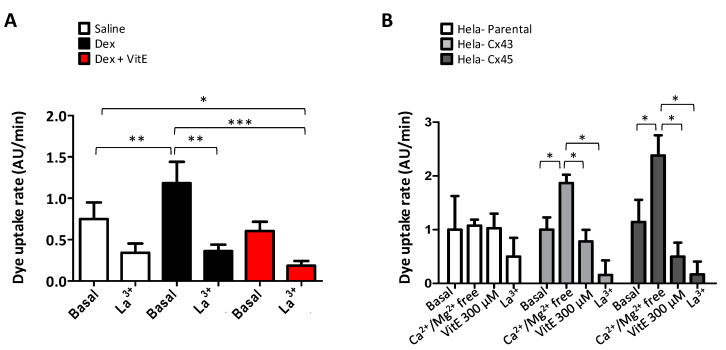
Vitamin E blocks the connexin hemichannel activity in skeltal myofibers of dexamethasone-treated mice and in transfected HeLa Cx43EGF cells. The activity of connexin hemichannels (Cx HCs) was measured using the dye uptake assays. (**A**) Etd^+^ uptake rate in isolated myofibers from the flexor digitorum brevis muscle of mice treated or not treated with (Dex, 10 mg/kg/day) or Dex+ VitE (*n* = 4). (**B**) Etd^+^ uptake rate of HeLa-Parental, HeLa Cx43 and HeLa Cx45 cells measured under control conditions (basal) and after exposure to (Ca^2+^/Mg^2+^ free) Krebs solution, and then treated with 300 μM VitE and lanthanum ion (La^3+^), a Cx HC blocker. Notice the enhanced dye uptake after incubation in Ca^2+^/Mg^2+^-free solution that induces opening of Cx HCs. *n* = 4 independent experiments for each cell line. * *p* < 0.05; ** *p* < 0.01; *** *p* < 0.001; as indicated with brackets.

**Figure 6 ijms-21-04094-f006:**
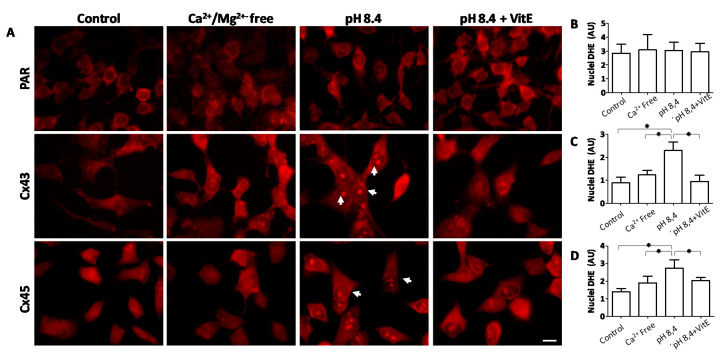
Ca^2+^ enters into cytoplasm and increases ROS generation in transfected HeLa cells. To measure ROS generation, the oxidation of dihydroxy ethidium (DHE) was visualized in HeLa-Parental, HeLa Cx43 and HeLa Cx45 cells exposed to control conditions and after exposure to (Ca^2+^/Mg^2+^ free) Krebs solution, alkaline solution (Krebs 8.4 pH) and alkaline solution with 300 μM VitE. (**A**) Representative images of nuclear DHE fluorescence. White arrow-heads indicate DHE positive nuclei. (**B**–**D**) Graphs showing the number of DHE fluorescence nuclei that was quantified using Image J software. *n* = 3 independent experiments. (**B**) HeLa parental cells, (**C**) HeLa Cx43 cells and (**D**) HeLa Cx45 cells. Three images from three independent experiments were evaluated * *p* < 0.05. Scale bar: 20 μm.

**Figure 7 ijms-21-04094-f007:**
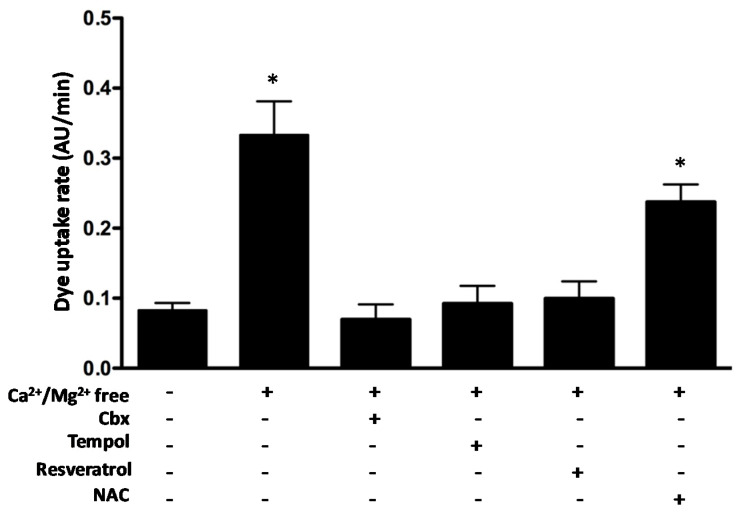
The antioxidant Tempol and Resveratrol blocks the Cx43 hemichannels in HeLa transfected cells bathed in divalent cation free solution. The activity of Cx43 HCs was measured using the dye uptake assays. Etd^+^ uptake rate of HeLa Cx43 cells measured under control conditions (first bar) and after exposure to (Ca^2+^/Mg^2+^ free) Krebs solution. Notice the enhanced dye uptake after incubation in Ca^2+^/Mg^2+^-free solution that induces opening of Cx HCs was prevented by carbenoxolone (Cbx; 100 µM) a Cx HC blocker. The cells were then treated with Tempol (500 μM), Resveratrol (200 μM) or N-acetylcysteine, NAC (5 mM). * *p* < 0.05 significantly different from basal conditions. *n* = 4.

**Figure 8 ijms-21-04094-f008:**
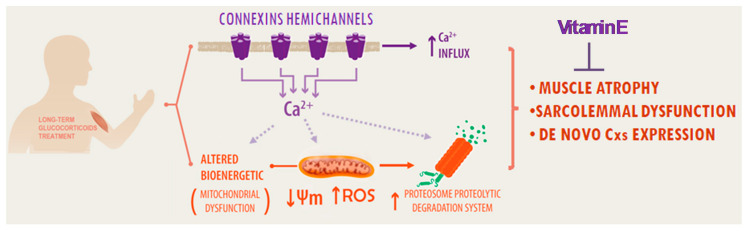
Dexamethasone treatment is associated with muscle atrophy. Our results suggest that these effects are explained by the expression and activation of connexin hemichannels in the sarcolemma, leading to increase the Ca^2+^ influx that reduces the mitochondrial membrane potential (Ψm), promoting mitochondrial dysfunction and increasing reactive oxidative species (ROS). Furthermore, a Ca^2+^-dependent pathway leads to activation of the proteasome proteolytic degradation system. These phenomena could be prevented by connexin hemichannel inhibition with vitamin E.

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
