# Peer review of "Vitamin E Blocks Connexin Hemichannels and Prevents Deleterious Effects of Glucocorticoid Treatment on Skeletal Muscles"

_ijms, 2020, doi:10.3390/ijms21114094_

Round 1

Reviewer 1 Report

Long-term corticosteroid (Dex) treatment induces muscle atrophy through incompletely understood mechanisms.  Reactive oxygen species and connexin expression/activity are likely implicated.  Thus the authors set out to elucidate the effects of an antioxidant (vitamin E) on ROS production, mitochondrial dysfunction and Cx signaling in a mouse model of Dex-induced muscle atrophy.

Major

  1. In the introduction, the link between Cxs, Dex, and oxidant stress is not very clear. What mechanistic links exist between atrophy, Cx, and ROS signaling, and what exactly is trying to be learned from the current investigation?  As written, the introduction is a series of seemingly unrelated paragraphs.  The authors should make clear what’s already known/reported (especially in their previous work, as reference 4) and what still needs to be answered.
  2. Page 2, line 48: how is loss of body weight indicative of muscle loss? It certainly contributes to and/or may be associated with muscle loss, but body weight changes alone do not specifically report changes in muscle mass.  can the authors report muscle weights?
  3. Why not include a Dex+VitE group in the Cx43 and Cx45 fl/fl mice? It seems like this could help answer some of the directional and mechanistic hypotheses posed in the discussion.
  4. This manuscript would really benefit from a summary figure to pull together all the key molecules/mechanisms. As is, the discussion is b it hard to follow.

Minor

  1. Is 7 days really “long term”? I suggest removing long-term from title, as this suggests a magnitude of ~ months
  2. Figure 4 is missing sample size per group
  3. In the discussion, pg 10, line 31-34: to me, if VitE prevented the generation of ROS by inhibition Cx expression/activity, then there needs to be evidence (at least from the literature) that Cx produce ROS. There’s no mention of this in the introduction.  To me, it seems without this, it’s much more likely that VitE simply scavenges ROS.
  4. Figure 1A needs to be analyzed by repeated measures ANOVA. The statistical markers then need to clarify which groups are being compared at which times.  As is, it’s not clear what’s being compared and is/is not significant.
  5. Each of the assessments (oxygen consumption, histology, mitochondrial function) were done in different muscles- soleus, FDB and TA. Does this have any impact on results/outcomes?
  6. Is an n=3 DEX+VitE sufficient? The authors should at very minimum acknowledge this limitation.  Better would be to increase to n=5 like the rest of the groups
  7. The temporal question in figure 2 is misleading. Rather what data from Cx43/45 deficient mice suggest is that Cx43/45 are required for Dex-induced ROS generation
  8. The authors might consider presenting Figure 4 earlier- perhaps around Figure 1. If Dex doesn’t induce Cx43/45 expression, then why bother with using genetic models of Cx deletion?
  9. While I like the idea to include other antioxidants besides VitE (tempol, NAC, resveratrol), I found the way in which the authors did so (non-muscle cell culture) to be underwhelming.

Author Response

ANSWERS TO REVIEWERS' COMMENTS:

We appreciate the positive comments from the reviewers and are thankful for the suggestions, which have certainly contributed to the improvement of our manuscript.

We have addressed the comments point by point. Please find our answers in italics after each comment.

Reviewer #1

Major

  1. In the introduction, the link between Cxs, Dex, and oxidant stress is not very clear. What mechanistic links exist between atrophy, Cx, and ROS signaling, and what exactly is trying to be learned from the current investigation?  As written, the introduction is a series of seemingly unrelated paragraphs.  The authors should make clear what’s already known/reported (especially in their previous work, as reference 4) and what still needs to be answered.

Answer: Responding to this concern of the reviewer, we included new information and references in page 2 line 25 to 34 and we also included a phrase with a brief hypothesis in page 2 line 44 and 45.

  1. Page 2, line 48: how is loss of body weight indicative of muscle loss? It certainly contributes to and/or may be associated with muscle loss, but body weight changes alone do not specifically report changes in muscle mass.  Can the authors report muscle weights?

Answer: We agree with the reviewer that changes in body weight alone do not specifically report changes in muscle mass. However, in this case it is associated to many other changes that we described in other figures plus data already reported in a previous article (PMID:27437607). Moreover, in this new version we included a supplementary figure 1 that shows the Gs muscle weight. We did not include this figure as part of figure 1 due to the low number of experiments in some groups. Unfortunately, we cannot repeat experiments because our laboratories have been closed up due to the COVID-19 pandemia and will remain closed for at least two to three more months.

  1. Why not include a Dex+VitE group in the Cx43 and Cx45 fl/fl mice? It seems like this could help answer some of the directional and mechanistic hypotheses posed in the discussion.

Answer: We did not include it since neither Dex nor Dex+VitE produce changes compared to saline in the analyzed parameters in Cx43/45 deficient mice. We also have a low n in that group and currently we cannot carry out any more experiments. We include in supplementary figure 1 and 2 some data in Dex treated Cx43/45 mice

  1. This manuscript would really benefit from a summary figure to pull together all the key molecules/mechanisms. As is, the discussion is bit hard to follow.

Answer: Thank you very much for you suggestion. In this new version of the manuscript in page 13, we included a new figure (Figure 8) that summarizes of our main findings.

Minor

  1. Is 7 days really “long term”? I suggest removing long-term from title, as this suggests a magnitude of ~ months

Answer: We appreciate that the reviewer has suggested us to change the title. In this new version of the article the title was changed as follows: “Vitamin E blocks connexin hemichannels and prevents deleterious effects of glucocorticoid treatment on skeletal muscles”.

  1. Figure 4 is missing sample size per group

Answer: : We apologize for this mistake and we have now included for Figure 2 (figure 4 in the previous version) the description of sample size in the figure legend (page 5 line 1 and 2 )and in the Materials and Methods section page 14 line 19

  1. In the discussion, pg 10, line 31-34: to me, if VitE prevented the generation of ROS by inhibition Cx expression/activity, then there needs to be evidence (at least from the literature) that Cx produce ROS. There’s no mention of this in the introduction.  To me, it seems without this, it’s much more likely that VitE simply scavenges ROS.

  • Answer: There is evidence that a gain function of Cx HCs formed by a mutated Cx induces oxidant stress (PMID:22393412). However and to our knowledge there is no direct evidence that HCs constituted by normal Cxs produce ROS. However, we have seen that cells that express Cxs 43 and 45 and are exposed to conditions that open them, induce oxidative stress (DHE (Fig. 6 )). In this new version of the article we have included in the introduction (page 2 line 25 to 34 ) some data related to the link between Cx HCs expression and oxidative stress.

  1. Figure 1A needs to be analyzed by repeated measures ANOVA. The statistical markers then need to clarify which groups are being compared at which times.  As is, it’s not clear what’s being compared and is/is not significant.

Answer: We apologize for this mistake and we have corrected this in figure 1 and in the corresponding figure legend, page 4 line 3 and 4. 

  1. Each of the assessments (oxygen consumption, histology, mitochondrial function) were done in different muscles- soleus, FDB and TA. Does this have any impact on results/outcomes?

Answer: We used different muscles in order to reduce the number of animals; also we selected different types of muscles for each measurement depending on the methodology and management. For example, it is easier to obtain intact fibers for microscopy of the FDB muscle. For OCR measurement, we selected the soleus (oxidative) over the TA (predominantly glycolytic) for its metabolic characteristics. For histology, we selected the TA muscle for its size and much easier for obtaining cryostat cuts.

  1. Is an n=3 DEX+VitE sufficient? The authors should at very minimum acknowledge this limitation.  Better would be to increase to n=5 like the rest of the groups

Answer: We agree with the reviewer that a higher n would be ideal, but for immunofluorescence, as for histology, we consider that n = 3 of muscles obtained from independent animals is a valid result since is possible to apply statistical analysis and validate the significance. Unfortunately, due to the contingency of the covid-19 pandemic, it is impossible for us to carry out new experiments to increase the number of animals.

  1. The temporal question in figure 2 is misleading. Rather what data from Cx43/45 deficient mice suggest is that Cx43/45 are required for Dex-induced ROS generation. The authors might consider presenting Figure 4 earlier- perhaps around Figure 1. If Dex doesn’t induce Cx43/45 expression, then why bother with using genetic models of Cx deletion?

Answer:We appreciate that the reviewer has suggested changing the presentation of the data. In this new version of the article we reordered the figures and figure 4 is now figure 2.

  1. While I like the idea to include other antioxidants besides VitE (tempol, NAC, resveratrol), I found the way in which the authors did so (non-muscle cell culture) to be underwhelming.

Answer:We agree with the reviewer that it would have been ideal to show results in muscle-type cells, however due to methodological limitations, we only had stable clones of Cx43-transfected HeLa cells.

Reviewer 2 Report

The manuscript by Balboa and coworkers analyzes the effects of VitE treatment on dexamethasone (Dex)-induced muscular atrophy and explores the mechanisms involved. Authors conclude that Dex induces an enhanced expression of Cx43/Cx45 at the cell membrane, which in turn would increase intracellular calcium concentrations, inducing mitochondrial dysfunction and ROS production, driving to muscular atrophy. VitE would antagonize these changes by directly inhibiting hemichannel function but also connexin expression.

This is a nicely designed and conducted experimental study that may have important clinical implications. Methods are correct and appropriate. However, some issues should be addressed before publication.

Major comments:

  1. Authors suggest that Dex effects on muscular atrophy are due to enhanced expression and activity of connexin hemichannels. If this is true, animals lacking Cx43/Cx45 should not develop muscular atrophy in response to Dex (or at least, they should develop an attenuated atrophy). I suggest to include data on body weight, cross-sectional myofiber area and atrogin expression in their Cx43/Cx45 expression-deficient mice (Cx43fl/flCx45fl/fl:Myo-Cre) with and without Dex treatment.
  2. It is not clear how VitE influences connexin to attenuate the effects of Dex on muscular atrophy. Is it due to changes in expression or in hemichannel function?. I suggest to clarify this by suggesting some experiments:
    • First, to analyze mRNA expression for Cx43/Cx45. During the discussion, authors suggest that Dex may increase mRNA expression for Cx43/Cx45, and VitE may attenuate this effect. However, this was not analyzed.
    • Second, have authors analyzed if an altered connexin translocation from internal locations is present?
    • Third, to confirm a direct action of VitE on connexin43/Cx45 function, I would suggest to include some patch clamp experiments, analyzing hemichannel conductivity after VitE treatment.
    • And last, authors suggest that intracellular calcium overload can be in part responsible for Dex effect, and that this was attenuated by Vit.E. However, they have not measured intracellular calcium. I suggest to include this data.
  3. Total expression of Cx43/Cx45 should be quantified by conventional western blot, in addition to show results on inmunofluorescence (figure 4). Also it is important to quantify connexin expression by WB in their Cx43/Cx45 expression-deficient mice (Cx43fl/flCx45fl/fl:Myo-Cre animals).

Minor comments:

  1. Why did authors select a concentration of Dex of 10 mg/kg?. Please specify.
  2. 1c (and other figures if it applies): data from a single animal should be averaged and presented as a single point. Statistical analysis should be performed also in that way.
  3. Some figure legends (figure 2, figure 3B, etc) have more * symbols (1, 2 or 3) than shown in the figures. Figure 3B-C does not show panel a or b.
  4. Figure 3D show lower values for OCR than those shown in wild-type animals. It is possible that a further reduction is not feasible and that this was the reason for lack of mitochondrial dysfunction in these animals. Authors should acknowledge this in the discussion, as a limitation of the study. This also seems to apply for mitochondrial membrane potential.
  5. Why ROS is seen as an augmented signal in the nucleus in figure 6, but not in figure 2?

Author Response

Reviewer #2

Major comments:

  1. Authors suggest that Dex effects on muscular atrophy are due to enhanced expression and activity of connexin hemichannels. If this is true, animals lacking Cx43/Cx45 should not develop muscular atrophy in response to Dex (or at least, they should develop an attenuated atrophy). I suggest to include data on body weight, cross-sectional myofiber area and atrogin expression in their Cx43/Cx45 expression-deficient mice (Cx43fl/flCx45fl/fl:Myo-Cre) with and without Dex treatment.

Answer:We agree with the reviewer and we did not show these data in the previous version of the article because this was published earlier (Cea LA, et al. Dexamethasone-induced muscular atrophy is mediated by functional expression of connexin-based hemichannels. Biochim Biophys Acta. 2016;1862(10):1891‐1899). In the new version of the article we have included a supplementary figure 2 with new data in Cx43fl/flCx45fl/fl:Myo-Cre mice treated with dexamethasone (10 mg/kg)and we described this data in page 5 line 3-7)

  1. It is not clear how VitE influences connexin to attenuate the effects of Dex on muscular atrophy. Is it due to changes in expression or in hemichannel function?. I suggest to clarify this by suggesting some experiments:
    • First, to analyze mRNA expression for Cx43/Cx45. During the discussion, authors suggest that Dex may increase mRNA expression for Cx43/Cx45, and VitE may attenuate this effect. However, this was not analyzed.
    • Second, have authors analyzed if an altered connexin translocation from internal locations is present?
    • Third, to confirm a direct action of VitE on connexin43/Cx45 function, I would suggest to include some patch clamp experiments, analyzing hemichannel conductivity after VitE treatment.
    • And last, authors suggest that intracellular calcium overload can be in part responsible for Dex effect, and that this was attenuated by Vit.E. However, they have not measured intracellular calcium. I suggest to include this data.
  2. Total expression of Cx43/Cx45 should be quantified by conventional western blot, in addition to show results on inmunofluorescence (figure 4). Also it is important to quantify connexin expression by WB in their Cx43/Cx45 expression-deficient mice (Cx43fl/flCx45fl/fl:Myo-Cre animals).

Answer: We appreciate that the reviewer has suggested experiments, but currently and indefinitely we will not be able to do experiments due to the pandemic. However, in figure 2 it can be seen that vitE decreases the expression levels of Cx43 and Cx45 in the membrane. In addition, in the dye uptake experiments (figure 5), it can be seen that vitamin E (acutely applied) also decreases dye uptake, so it would also decrease the activity of the Cx HCs.

Minor comments:

  1. Why did authors select a concentration of Dex of 10 mg/kg?. Please specify.

Answer: We chose this dose based on previous results obtained in our laboratory and data reported in the literature.  High-dose (10 a 25 mg/Kg) of dexamethasone treatment is a standard animal model of skeletal muscle atrophy. Furthermore these dose of dexamethasone has been show to induce atrophy, in part by the indirect activation of atrogenes . (Qin J, et al. Res Vet Sci. 2013;94(1):84‐89; Jesinkey S, et al. J Pharmacol Exp Ther. 2014;351(3):663‐673.)

  1. 1c (and other figures if it applies): data from a single animal should be averaged and presented as a single point. Statistical analysis should be performed also in that way.

Answer:We agree with the reviewer and we have change the figure 1C accordingly.

  1. Some figure legends (figure 2, figure 3B, etc) have more * symbols (1, 2 or 3) than shown in the figures. Figure 3B-C does not show panel a or b.

Answer: We apologize for this mistake and we have corrected this in the figure 2 and 4 (previous figure 3).

  1. Figure 3D show lower values for OCR than those shown in wild-type animals. It is possible that a further reduction is not feasible and that this was the reason for lack of mitochondrial dysfunction in these animals. Authors should acknowledge this in the discussion, as a limitation of the study. This also seems to apply for mitochondrial membrane potential.

Answer: We reviewed the literature and the OCR ranges in the units reported by us (pmol/(s*mg)) are within the ranges reported by previous studies. We believe that the differences are due to differences in genetic background of the wild type mice and Cx43fl flCx45fl/fl: Myo-Cre mice. In the new version of the article we have included a paragraph in the results section, page 6 line 44-47

  1. Why ROS is seen as an augmented signal in the nucleus in figure 6, but not in figure 2?

Answer: We disagree with the reviewer, since in figure 3 (previous figure 2), wild type mice treated with dexamethasone shows an increase in DHE red fluorescence in the nuclei of muscle fibers. This result is also represented in the graph showed at the top right. Moreover, Cx43/Cx45 expression deficient myofibers do not present this change on DHE in the nuclei (Bottom right graph).

Round 2

Reviewer 2 Report

The manuscript by Balboa and coworkers has been substantially improved and authors have successfully answered most of my comments.